# A Review: Application and Implementation of Optic Fibre Sensors for Gas Detection

**DOI:** 10.3390/s21206755

**Published:** 2021-10-12

**Authors:** Thomas Allsop, Ronald Neal

**Affiliations:** 1School of Engineering and Computer Science, University of Hull, Hull HU6 7RX, UK; 2Aston Institute of Photonic Technologies (AIPT), Aston University, Aston Triangle, Birmingham B4 7ET, UK; 3School of Engineering, Computing and Mathematics, University of Plymouth, Plymouth PL4 8AA, UK; r.neal@plymouth.ac.uk

**Keywords:** optical fiber sensors, gas sensing, gratings, surface plasmon resonances

## Abstract

At the present time, there are major concerns regarding global warming and the possible catastrophic influence of greenhouse gases on climate change has spurred the research community to investigate and develop new gas-sensing methods and devices for remote and continuous sensing. Furthermore, there are a myriad of workplaces, such as petrochemical and pharmacological industries, where reliable remote gas tests are needed so that operatives have a safe working environment. The authors have concentrated their efforts on optical fibre sensing of gases, as we became aware of their increasing range of applications. Optical fibre gas sensors are capable of remote sensing, working in various environments, and have the potential to outperform conventional metal oxide semiconductor (MOS) gas sensors. Researchers are studying a number of configurations and mechanisms to detect specific gases and ways to enhance their performances. Evidence is growing that optical fibre gas sensors are superior in a number of ways, and are likely to replace MOS gas sensors in some application areas. All sensors use a transducer to produce chemical selectivity by means of an overlay coating material that yields a binding reaction. A number of different structural designs have been, and are, under investigation. Examples include tilted Bragg gratings and long period gratings embedded in optical fibres, as well as surface plasmon resonance and intra-cavity absorption. The authors believe that a review of optical fibre gas sensing is now timely and appropriate, as it will assist current researchers and encourage research into new photonic methods and techniques.

## 1. Introduction

The first studies using optical fibres as sensory devices for structural health monitoring were published in the early 1970s [1,2]. Over the last five decades, research into sensory-device development, using optical fibres, has expanded significantly, covering a wide range of applications. Research into optical fibre gas sensing started in the 1980s, as it became apparent that these sensors could be readily accessed, remotely monitored in real time, and had multiplexing capabilities. In addition, they have the potential to operate in hazardous and extreme environments and are intrinsically safe in inflammable atmospheres as they are non-electrical; thus, spark generation is non-existent. The gases to be detected can be colourless, odourless, explosive, asphyxiating, and lighter than air. A great deal of effort has been devoted to the detection of methane [3], because of its relevance to the petrochemical and coal-mining industries [4]. These initial studies focussed on using optical fibre as an emitter and detector, with a cavity to measure absorption lines of methane [3]. More recently, methane detection has become a subject of interest because it is a greenhouse gas and has overtaken carbon dioxide detection because of its more significant contribution to climate change [5]. Gas sensing using optical fibres has commercial implications. During oil refinery turn-around and inspections, plant operators still use hand-held gas testing equipment in enclosed spaces or vessels to ensure safe working conditions for inspection. This puts the operator’s life at risk and it takes time to plan and to conduct gas tests. From a financial perspective, time under these set of circumstances is a very valuable commodity and saving time helps to reduce overall plant downtime, thus increasing productivity and profit [6]. 

Optical fibre gas sensors are still being actively researched because of their advantages and superior performance over non-fibre sensors, such as metal oxide semiconductor (MOS) and spectroscopic techniques. MOS sensors have a number of advantages over spectroscopic methods, which rely on measurements of absorption spectra [7], namely lower fabrication costs, miniaturisation, integration, and multiplexing capabilities. The downside of MOS sensors are problems and issues related to stability and chemical selectivity, along with their need to operate above ambient temperature, and, as a result of their electrical operation, there is the problem of spark hazards. Elevated temperature operation and sparking hazards are significant problems. Furthermore, MOS sensors have issues with relatively low sensitivities and their performance is sensitive to environmental factors [8,9]. Spectroscopic techniques need appropriate apparatus to measure optical absorption, emissions, or scattering spectra [10], and the gas must have a significant and distinct response within the optical spectrum being used. A limitation of the spectroscopic technique comes from the limited number of light sources available to illuminate the gas samples.

The other technique used in laboratories is gas chromatography (GC), which is very accurate and can be used for very specific gases and detecting their concentrations [11,12]. The issues with GC are that it is not easy to translate laboratory procedures into use as an in-line, remote gas sampling and detection scheme at a site or plant. In addition to the aforementioned gas sensing techniques, there are other techniques, such as photoacoustic spectroscopy for trace gas sensing [13,14], laser photoacoustic spectroscopy [15], and electronic noses [16]. In fact, there are myriad of detection and sensing schemes for various gases [17,18].

This review will focus on optical fibre sensor groups that have been researched and investigated. There are many types of fibre optic sensors that are being developed and studied at present, but we will classify the sensors into several groups, listed as follows: Group 1—optical fibre spectroscopic sensors, consisting of intra cavity, external cavity, photonic crystal fibre (PCF) and surface enhanced Raman spectroscopy (SERS) [19,20,21,22,23,24,25,26]. Group 2—fibre grating sensors, including long period gratings, Bragg gratings, and tilted Bragg gratings [27,28,29,30,31,32]. Group 3—evanescent field sensors, including in-fiber Mach-Zehnders, exposed fibre cores, conically tapered fibres, and random-hole optical fibers [33,34,35,36,37,38]. Group 4—plasmonic sensors, including surface plasmon resonance, localised surface plasmons, and conjoined surface plasmon [39,40,41,42,43,44]. 

We will explain the underlying physical mechanisms being exploited, and, where possible, make performance comparisons of spectral sensitivity, limit of detection, range of operation, and response time. Some optical fibre sensors require a transducer, for example thin film coatings, to provide chemical selectivity. Examples of materials in use will also be discussed. The authors hope that this review will be helpful to the existing research community and encourage new researchers to participate.

## 2. The Classification and Mechanism of Sensors

### 2.1. Spectroscopic Optical Fibre Sensors

Generally, spectroscopic techniques have been applied to fibre-optics sensors and are relatively successful in gas sensing applications. Two major mechanisms underpin these types of sensors. The first utilises fairly standard spectroscopic techniques, in which the gas absorbs incident optical radiation at specific wavelengths [19,20]. The second is a direct interaction of the evanescent field that exists at the boundary interface between the waveguide (supporting guided, leaky mode, or both) and the surrounding medium, in this case, the optical fibre, the waveguide, and the surrounding medium of the gas. This interaction can be enhanced by confining the guided mode to a reduced physical volume of the fibre. This restriction spatially extends the evanescent field into the surrounding media (the gas), thereby increasing the sensitivity [45]. 

Thus, these types of optical fibre gas sensors needs to have a “chamber” or cavity into which a sample of gas can be placed and then analysed. A gas sensing intra-cavity can be fabricated within the fibre [19,21] or as a separate external cavity [20]. Examples of the former use photonic crystal fibre (PCF), in which holes within the fibre constitute the cavities [23,24]. Some examples are illustrated in Figure 1a–d. The “grapefruit” fibre in Figure 1a has a core with a very small diameter. The guided mode permeates into the air gaps. Figure 1e shows a capillary, hollow optical fibre that has a gas inlet and outlet facility, spliced onto a standard single-mode fibre (SMF) [46]. This fabrication constitutes the classical spectroscopy format, in which gas is illuminated and the resulting spectra is detected and analysed. A further example involves micro-machining a section of a SMF to create a cavity in close proximity to the core of the fibre [47,48]. Structuring the cavity involves femtosecond-inscription followed by HF etching, as shown in Figure 1f. 

The second class of sensors in this category is classified as surface Raman spectroscopy (SRS), and there are a number of variants [49,50,51]. The underlying principle of operation is inelastic scattering of incident photons by the gas, some of which induce excitation of vibration states of gas molecules generating phonons and detectable energy dissipation. Other wavelengths of the incident spectra are reflected, giving rise to further Raman scattering (infrared absorption) [49]. Resonance Raman spectroscopy (RRS) occurs when the incident photonic energies match the electronic–vibrational–excitation states of a gas. This matching of energies can lead to an increase in the intensity of Raman scattering, which has found application in low-level chemical-compound studies. Finally, in this section, we include surface enhanced Raman spectroscopy (SERS), as illustrated in Figure 1g. The interaction between the incident optical energy and gas occurs on a flat, nanostructured, metalized surface, fabricated in the fibre. The exact interplay between the incident optical energy and the gas is currently being researched in an effort to fully understand the mechanisms involved. Evidence suggesting that the irradiance excites localised surface plasmons (LSPS) on the metalized surface is growing. Evidence also indicates that the plasmons are sensitive to the shapes, sizes, and orientations of surface irregularities and contribute to the conversion of optical energy (photons) to mechanical energy (phonons) and, hence, to Raman scattering. Varying the surface pattern changes the SERS response, meaning that the device is suitable for both chemical and gas sensing. 

### 2.2. Optical Fibre Grating Sensors

Grating sensors are classified as follows, fibre Bragg grating (FBG), tilted fibre Bragg grating (TFBG), and long period grating (LPG) [27,28,29,30,31,32]. Research into fibre grating sensing has been extensive, resulting in a substantial number of publications. Because of this, the authors do not intend to provide a detailed analysis, but rather to provide a pathway for readers to obtain more information. All three sensors have a commonality, which is that they have a comb-like, periodic structural variation in the refraction index within the fibre core that induces a coupling action between the core mode and other modes supported by the fibre. There are a number of different fabrication methods in use; for example, ultraviolet (UV) phase-mask inscription, UV point to point, direct-write femto-second laser inscription, and fusion-arc [52,53,54,55]. The mode coupling mechanism depends on the type of grating, physical geometry, and the material used in the fabrication process, and the core mode can be guided, lossy/leaky, and radiative [56,57].

In brief, a fibre Bragg grating can be described as a radially symmetric refractive index perturbation, away from the central axis of the optical fibre, with a typical period of perturbation of approximately 1 μm in the core of the fibre. The forward propagation mode excites, via back-reflected light, counter-propagation core modes, including minority leaky modes and radiative modes [56,57], see Figure 2a. Although the core mode interacts with the surrounding medium, the interaction is not very strong and there are a number of fibre configurations developed to enhance the interaction. For example, exposing the core using intra-cavities or by means of material coatings. Pallidum, is such an example, and in the presence of hydrogen the material can physically change shape, thus creating a strain associated with the hydrogen that the FBG can detect, in the transmission or reflected spectrum [58]. 

Alternatively using an optical fibre that can create non- or adiabatic mode evolutions, such as biconical tapers, couples light to higher order modes that interact the surrounding more [59]. A key spectral property of a FBG is the narrow spectral bandwidth of transmission and reflection, resulting in a good resolution and limit of detection (LOD). Figure 2a shows a typical spectral response.

The issue of using FBGs is they are intrinsically insensitivity to changes in the surrounding medium, this can be seen in that the phase-matching condition for a FBG is the wavelength of resonants ($\lambda_{B}=2n_{eff}\Lambda$) and the maximum grating reflectivity ($R=tanh^{2}\left(\kappa L\right)$ where *n_eff_* is the effective index of the core mode, *L* is the period of the fibre Bragg grating, *L* is the length of the grating, and *k* is the coupling coefficient). In the case of sinusoidal uniform Bragg grating, the coupling strength between the forward and backward modes can be described by a coupling coefficient: κ=(πδnλB)η. The variable *η* represents the fraction of fibre mode power contained by the fibre’s core, and can be estimated using the normalised parameter *V*. On the basis that the grating is uniform, then *η* can be approximated by $\eta \approx 1-V^{-2}$ and using =(2πaλ0)ncore2−ncladding2, *a* is the core radius, *n_core_* is the core refractive index, *n_cladding_* is the cladding refractive index, and *λ*_0_ is the wavelength of the light in free space. Wavelength $\lambda_{B}$ is a function of $\lambda_{B}\left(\lambda ,\;\varepsilon ,\;T,\;S_{f}\right)$ with the phase condition $2n_{eff}\Lambda$ and should be written as $\lambda_{B}=2n_{eff}\left(\lambda ,\;\varepsilon ,\;T,\;S_{f}\right)\Lambda \left(\varepsilon ,\;T\right)$. The variables are wavelength (*l*), strain (*ε*), temperature (*T*), and *S_f_* is the waveguide geometry and material factor [57]. Therefore, the use of FBGs is dependent on affecting one of these variables. The general sensitivity of a conventional FBG, written in a step-index optical fibre, is 1.2 pm/μ*ε* [52]. A FBG coated with palladium can be used to detect hydrogen using the parameter *ε* [58]. However, another example is to remove the fibre cladding by polishing or etching to within a close proximity of the core so that the evanescent field of the core mode is exposed to the surrounding medium, changing the phased-matched condition of the FBG, using the parameter *S_f_*. 

In the case of tilted fibre Bragg grating (TFBG), the refractive index perturbation in the core of the TFBG is radially asymmetric from the central axis of the fibre. This increases the coupling to leaky and radiative modes [57,58], which, in turn, ameliorates exposure to the surrounding medium (the gas being tested). Using appropriate material coatings can enhance the linkage even more. Figure 2b shows an example of the transmission spectra of a TFBG. Other studies into TFBG behaviour have demonstrated another optical phenomenon; this being the generation of surface plasmons [60].

There are two major components in the transmission spectra of TFBGs; at a longer wavelength is the core mode Bragg wavelength phase matching condition and at a short wavelength the cladding mode resonance wavelength, Figure 2b. The Bragg wavelength can be expressed as $\lambda_{B}=\left(n_{eff\;core}+n_{effcore}\right)\Lambda$ where each *n_eff core_* represents the forward propagating core mode and counter-propagating core mode. The tilt angle of grating plane with respect to the fibre axis, the grating period along fibre axis, could be modified as Λg=Λcosξ where *ξ* represents the tilt angle and is defined as the angle from the perpendicular section of the optical fibre. Thus, for the Bragg wavelength condition (neff core(λ, ε, T, Sf)+neffcore(λ, ε, T, Sf))Λcosξ and the resonance wavelength of cladding mode is determined by (neff core(λ, ε, T, Sf)+nνeffcladding(λ, ε, T, Sf, ns))Λcosξ, where $n^{\nu }{}_{effcladding}$ is the effective refractive index of the νth cladding mode [61]. Therefore, the cladding mode is affected by the surrounding medium because the waveguide interface support the cladding mode is the cladding material and the refractive index of the surrounding medium. Thus, by functionalising (altering the permittivity) the out-layer/coating of a TFBG to the presence of gas, it is possible to create a TFBG optical fibre gas sensor [61].

Finally, in this section, we discuss the behaviour of an optical fibre that has a long period grating (LPG) written into its structure. An LPG, in an optical fibre, supports many cladding modes and thus the LPG induces a corresponding series of attenuation bands in the transmission spectrum of the fibre. The centre wavelength (*λ_υ_*) of an attenuation band is specified by the phase-matching conditions $\lambda_{\nu }=\delta n_{eff_{\nu }\;}\Lambda$ where $\delta n_{eff_{\nu }\;}=n_{eff\;core\;}\left(\lambda ,\;\varepsilon ,\;T,\;S_{f},n_{i}\right)-n^{1\nu }{}_{effcladding}\left(\lambda ,\;\varepsilon ,\;T,\;S_{f},\;n_{s},\;n_{i}\right)$ and clear defining of $n^{1\nu }{}_{effcladding}$ is the effective index of the *υ*th radial cladding mode, both indices being dependent on the indices of the various fiber layers, *n_i_,* and the wavelength *λ*. Additionally, $n_{cl}{}^{1\nu }$ is a function of the refractive index of the surrounding medium, *n_s_*. Λ is the period of the LPG, *T* the temperature, and *ε* the strain experienced by the fibre. The quantity $\delta n_{eff_{\nu }}$ is the differential effective index between the core and cladding modes. The superscripts denote the LP_01_ core mode and the HE_1*υ*_ axially symmetric cladding modes (from this point forward, for brevity we replace 1*υ* with *υ*). We are assuming here that the grating consists of a circularly symmetric index perturbation transverse to the axis of the fibre, so that the only non-zero coupling coefficients between the core mode and the cladding modes involve cladding modes of the azimuthal order 1 [62]. This creates a series of attenuation bands in the transmission spectra of the LPG, Figure 2c. Again, similar to TFBG the cladding modes show spectral sensitivity to the surrounding medium, but here the sensitivity can be significantly higher. The relationship between the cladding mode index and the surrounding medium and the analytical expressions are given elsewhere [28]. 

### 2.3. Evanescent Field Sensors

In this section of the review, we discuss the development of devices that exploit the properties of evanescent electric fields without the use of metallic coatings or gratings that interact with the surrounding environment. There are four principal types, namely, in-fibre Mach-Zehnder (MZ), exposed fibre core, conically tapered fibres, and random-hole fibres [33,34,35,36,37,38]. A number of different methods have been investigated to fabricate fibre MZ interferometers; for example, using a fibre fusion splicer to shape a pair of biconical fibres or to create a fibre with a large lateral off-set, or fusion splicing sections of a PCF. Another method is to use a fusion splicer section of optical fibre with a thin core compared to the fibre it is being spliced into [63]. One pathway has a constant index as a result of not being exposed to the surrounding media, whereas the other pathway is exposed, changing its index. This causes a phase difference and, hence, interference fringes are formed on recombination. The combined light intensity transmission spectrum of in-line fibre MZ is given by I(λ)=I1+I2+2I1I2cos[2π(ncon−ns)L0λ+ϕ0]; where *I*_1_ and *I*_2_ are the light intensity along two optical paths, as shown in Figure 3a. *λ* is the free-space optical wavelength in air, *L*_0_ is the geometry length of the MZI (length of the sandwiched SMF), *ϕ*_0_ is the initial phase of the interference, *n_con_* is the refractive index of the control path (in many cases, the index of the core on SMF) and *n_s_* is the refractive index of the sample in the MZI cavity. Destructive interference can be obtained when the phase difference between the control path (centre core mode) and the sensing path (ns) equals (2m + 1) *π*, where m is an integer. Therefore, the wavelength can be expressed as in the destructive interference occurs λm=2ΔneffL02m+1 with $\Delta n_{eff}=n_{con}-n_{s}$. This yields a free spectral range of FSR=λ2ΔneffL0. Thus, as *n_s_* increases then the interference pattern shifts, as in Figure 3b. 

Two devices have been developed in which the cores of the optical fibres are exposed directly to the surrounding media. The first makes use of so-called D-fibre, in which the cladding has been polished or etched away [64]. The second uses micro/nanofibres, see Figure 3c,d. In both devices, exposure allows access to the core mode boundary conditions by the surrounding media, which in turn alters the optical characteristics, such as the attenuation coefficient, propagation constant, and dispersion [65,66]. In the micro/nano device, the core mode electric field extends further into the surrounding media, although causing a higher attenuation improves the sensitivity [67]. Typical diameters of these micro/nanofibers are approximately 3 μm or less. Various parameters govern the spectral behaviour of these micro/nanofibres, such as, the mode field intensity, spot size, central peak intensity evolution, and adiabaticity. These parameters can all be calculated and the authors will not go into too much detail here as this can be found elsewhere.

The authors do not intend to delve further into mathematical analyses of these devices other than to indicate that a description of the fundamental mode behaviour can be obtained from three different sets of eigenvalue equations for various diameters. This is done numerically: firstly, for fibre diameters from 62.5 μm to 30 μm, using the weakly guided mode equation $u\frac{J_{\nu +1}\left(u\right)}{J_{\nu }\left(u\right)}=w1\frac{K_{\nu +1}\left(w1\right)}{K_{\nu }\left(w1\right)}$; where *u* and *w* are modal parameters defined as $u=a{\left(k^{2}n_{1}{}^{2}-\beta^{2}\right)}^{1/2}$ and $w1=a{\left(\beta^{2}-k^{2}n_{1}{}^{2}\right)}^{1/2}$, *a* is the core radius, *k* is the wavenumber k=2πλ, *n*_1_ is the index of core, *β* is the mode propagation constant, *J_υ_* is the Bessel function of the first kind of order *υ*, and *K_υ_* is the modified Bessel function of the second kind of order *υ* [68]. Secondly, for fibre diameters from 30 μm to 2 μm, the modal field is determined by using weakly guided approximation for doubly clad fibers with a core index higher than the inner cladding index. This is done numerical, solving the weakly guided dispersion equation; the authors refer the reader to [69]. Thirdly, from a fibre with a diameter of 2 μm to sub-μm, the modal field can be determined by the exact mode eigenvalue equation for the step-index profile optical fiber $\left(\frac{{J^{\prime }}_{\nu }\left(u\right)}{uJ_{\nu }\left(u\right)}+\frac{{K^{\prime }}_{\nu }\left(w\right)}{wK_{\nu }\left(w\right)}\right)\left(\frac{{J^{\prime }}_{\nu }\left(u\right)}{uJ_{\nu }\left(u\right)}+\frac{n_{3}{}^{2}}{n_{2}{}^{2}}\frac{{K^{\prime }}_{\nu }\left(w\right)}{wK_{\nu }\left(w\right)}\right)={\left(\frac{\upsilon \beta }{kn_{2}}\right)}^{2}{\left(\frac{V}{uw}\right)}^{4}$; where $w=a{\left(\beta^{2}-k^{2}n_{2}{}^{2}\right)}^{1/2}$ and $V^{2}=u^{2}+w^{2}$ and *n*_3_ is the index of the surrounding environment [70]. 

The behaviour of the third group of sensors under this heading is similar to those using micro/nano fibres. The coupling from the core mode to the cladding modes in the tapers is determined by the geometry of the tapers. The propagating radiation in the core is coupled into higher order modes that have a larger proportion of modal field extending into the surrounding media. Typical geometry and spectral responses are shown in Figure 3e. 

Mode coupling in tapers is investigated using the slowness criteria [70], which determines whether a taper is adiabatic or non-adiabatic. The magnitude of the gradient of the radius along the taper needs to be smaller than the adiabatic length scale [71] so that a strong mode coupling occurs [70,71]. The strength of a couple for the taper is governed by |∂r∂z|≪rzb with zb=2πβ1−β2, see Figure 3f. Here, *z_b_* is the beat length between two modes with propagation constants of *β*_1_ and *β*_2,_ and *r* is the radius of the fiber along the taper, rzb is the adiabatic length-scale criteria. The method to evaluate a taper is through using the finite element method [72]. The tapers are fabricated using several methods, such as a fusion splicer or the flame method with a controlled pulling mechanism, or specific equipment like automated glass processor workstations, such as Vytran or FiberBridge Photonics [73,74,75,76].

Finally, we discuss the construction and behaviour of a fibre evanescent sensor defined as a random-hole fibre [77]. As shown in Figure 3g, the fibre has a collection of holes, randomly distributed along the axis of the fibre. The holes allow a gas to interact with the evanescent field of the guided modes. Absorption occurs in the infrared part of the spectrum because of the vibrational-rotation of specific bonds within a molecule. The higher the molecular concentration (M), the greater the absorption. Lambert’s law describes the absorption process, $I=I_{0}\exp\left(-\frac{2\alpha_{0}}{M}\right)$, where *α*_0_ is the absorption coefficient of the gas, determined by the gas species and the interrogating wavelength [38]. This approach to gas sensing is still in its infancy as a research topic. 

### 2.4. Plasmonic Sensors

A plasmonic sensor is a sensor based on the generation of a surface plasmon. A surface plasmon is a type of light that exists at a metal–dielectric interface; electrons are excited by an illuminating irradiance, which creates an E-field that travels along the surface of the metal [78]. The parameters that govern the spectral behaviour and optical characteristics and the sensing performance of surface plasmons, include refractive index and extinction coefficient of the metal (or permittivity), dispersion of metal, the thickness of the metal, surface roughness of the metal and the surface topology of metal, as well as the refractive index of the surrounding medium that creates the interface [78,79]. These various conditions lead to various kinds of operations of surface plasmons. Firstly, the long-range surface plasmon and the short-range surface plasmon, which are observed in transmission spectra through surface plasmon resonance (SPR). SPR is an attenuation band when there is a phase-matched condition of illuminating light and the surface plasmon (SP) of a metal surface. This type of surface plasmon transverses a plane of a metal surface [78]. These surfaces are fabricated using a variety of techniques, such as vacuum deposition, RF sputtering, and chemical vapor deposition [80,81,82]. Secondly, are the localized surface plasmons (LSP), associated with nanostructured or nano-patterned metal surfaces with an SP with an electric field in close proximity to the surface of the metal surface, thus they exist only where the metal exists. The increase in interest and use of LSPs has come about due to the improvement of fabrication techniques of nano-patterning of the top of metal surfaces [40,83]. The excitation of a single LSP is strongly dependent on the shape and size distribution of the nano-patterned metal on a supporting surface topology. To increase the LSP spectral shift, various types of nano-pattern arrays, such as nano-spheres, nano-wells, and nano-antenna, have been fabricated via either chemical or laser lithography [83,84].

The conventional SP the propagation constant, $\beta_{SP}$, of the plasmon field is dependent on the permittivity (refractive index) of the metal coating and the media in contact with the metal coating. The dispersion relation for two homogeneous semi-infinite media [78,79] is $\beta_{SP}=k\sqrt{\frac{\varepsilon_{m}\left(\lambda \right)n_{s}{\left(\lambda \right)}^{2}}{\varepsilon_{m}\left(\lambda \right)+n_{s}{\left(\lambda \right)}^{2}}}$; where *k* is the free space wave number = 2π/λ, *ε_m_* is the dielectric constant of the metal (*ε_m_* = *ε_mr_*+ *i ε_mi_*), and *n_s_* is the refractive index of the dielectric sample to be tested. Using classical free-optics, the Otto and Kretschmann configuration of the output of the experimental apparatus is used to detect the resonance, which is produced in the reflection spectrum and the phase-matched condition is given by 2πλnpsin(θ)=Re[βSP], where *θ* is the angle of incident in the apparatus (the prism) and np is the refractive index of the prism; further details can be found elsewhere [78,79]. Using these two expressions, the propagation modes can be estimated for a given optical fibre, specific geometry, dispersion, and refractive index. There are several methods available for doing this depending on the complexity of the waveguide’s spatial geometry and the differing refractive indices involved. A general approach is to use conformal mapping techniques that allows simplification of the waveguide spatial geometry and then to apply Yeh’s algorithm method to account for the different index layers, with the outer most layer representing the surrounding medium. Using this detail, the various waveguide modes can be identified using the dispersion relationship that can be supported by the waveguide/optical fibre. Using the estimated dispersion relationship in conjunction with Fresnel’s equations for a layered system can yield a quantitative spectral description of the resonance condition of a SPR. The full details can be found elsewhere [79]. There are various optical fibre configurations, a typical SPR spectral transmission response to changes in the surrounding medium index is shown in Figure 4a. 

The second group within the classification of plasmonics is the localised surface plasmons (LSP), in this particular case, the surface plasmon is associated with the surface topology and the metallic artefacts on a surface, such as nano-spheres, nano-wells, and nano-antenna. Thus, LSP are associated and confined to individual or adjacent artefacts, an example of surface artefacts is shown in Figure 4b. Spatial extension into the surrounding medium is very small on the nanoscale, thus if specific chemical receptors are attached (immobilised) onto the surface metallic artefacts, the LSP evanescent will only/or majorly see the chemical receptors and their reactions, thus potentially yielding greater sensitivity for a specific chemical species. The calculation of LSP using a small/nanoscale metallic surface artefact or nanostructures is complicated. The optical properties of LSPs are determined by a series parameters: The size, shape and dielectric properties of the nanostructured materials [85,86,87]. This can be seen for nano-spheriod-shaped metallic artefacts. Assuming the size of a spheroid is significantly smaller than the wavelength of extinction, then the approximate condition *a/**λ*
*≤* 0.1 applies. The extinction spectrum generated by the metal composite spheroid as a function of wavelength is given by E(λ)=24πNAa3ε(λ)SR32λIn(10)[Im(ε(λ)ms)(Re(ε(λ)ms)+χε(λ)SR)2+Im(ε(λ)ms)2]; where *ε(λ)_ms_* is the complex dielectric function of the metal composite spheroids, *ε(λ)_SR_* is the complex dielectric function of the surrounding medium of the sensing platform, *a* is the mean radius of spheroids, and *N_A_* is the real density of the nano-blocks/spheroids [85,86]. The other variable is χ and takes into account the geometry of the spheroid and is solved analytically; χ can be defined as [85]: $\chi =-1-2{\left[\theta_{0}{}^{2}-\frac{\theta_{0}\left(\theta_{0}{}^{2}+1\right)}{2}\cos^{-1}\left(\frac{\theta_{0}{}^{2}-1}{\theta_{0}{}^{2}+1}\right)\right]}^{-1}$ variable *θ*_0_ is defined as θ0=(Ymean2Xmean2−1)−12. A myriad of optical fibre configurations have been investigated to create surface plasmons, both LSP and conventional SP, and have been used in the past and at present. Two such examples are shown in Figure 4c,d.

The third plasmonic fibre sensor is based on the strong coupling of adjacent LSP on neighbouring nano-antennae, or conjoined LSPs. These conjoined LSPs effectively create a conjoined surface plasmon [44] that arches over the array of nano-antennae; a typical array is shown in Figure 4e. This creates an evanescent field that is suspended above the majority of the nanostructured surface of the sensor, this can be seen from FEM modelling and the resultant associated E/H fields of the conjoined infrared surface plasmon are shown in Figure 4f. This creates a longer propagation length and interaction length producing ultra-high spectral sensitivities. This plasmonic device for detection systems is in its infancy at present, with only a few examples in use for gas sensing [88]. 

## 3. Gas Species Selectivity

We have thus far, discussed measurement of the refractive index of materials in general using an optical fibre sensor. The next step is to design a sensor to be chemically specific. In order to do this, we found that an additional transducing material was required that would react with the target chemical species and produce absorption line information for measurements. In this section, we describe various strategies used to achieve chemical selectivity. There are a myriad of gases and volatile materials that researchers and engineers are interested in studying for a number of reasons (ranging from safety issues to general analytical analysis). We decided to focus mainly on the greenhouse gases, methane, carbon dioxide and nitrous oxides, and water vapour. Greenhouse gases, of course, are linked to climate change, which is of major international concern [89,90,91].

### 3.1. Methane

Recently, methane has achieved much greater prominence in the public domain because of its more significant role in the greenhouse effect than carbon dioxide [92]. There is a substantial amount of published work using spectroscopic absorption relating to molecular transitions in near- and mid-infrared spectra, which is used in conjunction with a cavity, such as hollow fibres or micro-structured fibres [93,94,95]. Methane has infrared absorption bands, the *υ*_2_ + 2*υ*_3_ bands located at 1333 nm and the 2*υ*_3_ overtone band at 1666 nm, respectively. The *υ*_2_ + 2*υ*_3_ band of methane is a weak absorption line and there are additional measurement problems due to cross sensitivity with water vapour.

The absorption band at 1666 nm is quite strong, has minimum cross-sensitivity with other gases, and is a much stronger than the *υ*_2_+ 2*υ*_3_ band. The 1666 nm line is widely favoured for spectroscopic sensing. Other absorption wavelengths investigated were 1.33 μm, 1.53 μm, 1.65 μm, and 3.2 μm [93,94,95]. Major issues using spectroscopic detection techniques are long gas diffusion times, filling times (approximately one hour), and the apparatus is cumbersome and not easily made portable or robust. Furthermore, the spectral response of some of the used cavities can be complicated by the presence of fringe-like spectral features that can be observed in the output of sensors, which limit the accuracy of resolution and of detection [94,95].

Another major strategy for the detection of methane is chemisorption, a term used to describe a chemical reaction on a film surface, causing a change in the optical properties of the film. Typically, this strategy is used in conjunction with the other types of sensors (mentioned previously). Zinc oxides have a specific reaction with methane. Initially an oxygen molecule will attract an electron from the conduction band of ZnO and create O_2_−. This forms ZnO:O_2_− species on the surface of the ZnO; it is known that methane breaks down into C′H3 and ′H [96], thus, the hydrogen radial reacts with the O_2_− but has a high activation energy. Thus, these need to operate at temperatures in excess of 200 °C [97]. Lowering the activation energy (decreasing the Gibbs free energy) has been achieved using catalytic nodal metals, such as a palladium–silver-activated ZnO surface [98] or platinum [99]. This is in effect a redox reaction, resulting in a reduction in the permittivity of the ZnO [99]. Tin (IV) oxide (SnO_2_), an n-type semiconductor has been shown to alter the permittivity of methane as a result of the methane molecule acting as a reducing agent, donating electrons to the SnO_2_ film and using graphene to absorb the methane [100].

The use of methane-sensitive films containing cryptophane molecules has been demonstrated to be a useful technique for methane detection. These are organic supra molecular compounds containing large numbers of carbon atoms with structures of aromatic rings. Cryptophane molecules are shaped like a cage with the top and bottom of the cage made up of units of aromatic rings. The mid-section (the bridges) of the cryptophane consist of other organic molecules, which offer a variety of shapes, volumes, and chemical properties for the hydrophobic pocket inside the cage to be modified, making cryptophanes suitable for encapsulating many types of small molecules and even chemical reactions [101]. In particular, the cryptophane E molecule, which is a functional material that has a direct photosensitive property for methane, when a methane molecule enters the cage, the dipole moment of cryptophane molecule changes, creating changes in the permittivity and thus altering the refractive index of the cryptophane E molecule [101]. This approach is relatively new approach and shows promise. Work has been published using cryptophane E [102,103,104] with a sensor working at ambient temperatures and using cryptophane A has been selected as a methane-sensitive film [105].

Recently, coatings of graphene and/or carbon nanotubes (CNT), along with a polymer with an underlying nodal metal, have been used to make sensors specific for the detection of methane [106,107,108]. The outer coating is sometimes referred to as a nanocomposite where the graphene and/or CNTs yields a large surface to react with the surrounding environment and a polymer is used to produce a selective response to methane. The polymers reported to date include poly(methyl methacrylate) [106,107] and the use of a reduced oxide graphene [108]. Other polymers have also shown some success in chemical selectivity with methane, such as poly(acrylic acid)-carbon nanotubes/polypropylene amine hydrochloride working with a cryptophane molecule [109,110]. 

### 3.2. Carbon Dioxide

There has been a significant increase of spectroscopic techniques for the detection of carbon dioxide [111,112,113]. The principal absorption wavelengths are in the near and mid infrared, as follows: spectrum bands of lines are at ~2.005 μm (R branch), ~2.015 μm (P branch) [113] with absorption lines in the mid-infrared at ~4.2 μm [111,112] and ~2.6 μm [111]. There are some interference issues due to water vapour absorption lines in the range 2 μm and 6 μm causing cross-talk in the sensitivity and selectivity. Conventional SMF optical fibre suffers a loss in sensitivity at longer wavelengths, resulting in high attenuation. Optical fibres with a greater transparency at wavelengths from 1.5 μm to 10 µm have been developed, and are known as chalcogenide fibres [114]. This type of fibre has manufacturing problems, resulting in attenuation values of 0.1 to 0.5 dB/m, but have been found to expel water vapour if operated over 100 °C. 

An alternative method to spectroscopic techniques for the detection of CO_2_ is the use of an addition material with, or on, the optical fibre that has a specific reaction with CO_2_. A range of different materials have been reported in the literature over recent years, such as xerogels doped with 1-hydroxy-3,6,8-pyrenetrisulfonic acid trisodium salt (HPTS) [115], also known as pyranine. This CO_2_ sensor is based on pyranine, a pH-sensitive fluorescent indicator dye. The dye, in the presence of CO_2_ has an ion transfer that alters the absorption features of pyranine and xerogel at 396 nm and 460 nm, effectively reducing absorption at these wavelengths [115]. This reaction occurs at room temperature and has reasonable response times. Another approach uses oxidation or reduction reactions, where an electron transfer process occurs between a gas and a material. An example of this is hybrid nickel/reduced graphene oxide (NiO/rGO) which is a structured coating material. The reactionary wavelengths are 670 nm and 771 nm, and it is the combination of the nanostructured material and its chemical composition in the presence of CO_2_ that triggers the reaction, altering the distribution of the O-, O^2^^−^ and O_2-_ radicals within the materials, changing the electron density and hence the permittivity. The result is a small but significant change in the emission wavelengths [116]. A redox reaction using single-wall carbon nanotubes (CNTs) has been used for CO_2_ detection [117], the chemical selectivity is discussed in terms of the activation energy, which has been lowered to allow for room temperature operation. It is known that N_2_ can act as a redox agent for the CNTs, but the activation energy of this reaction is high; thus, elevated temperatures in excess of 500 °C are required [118,119].

Yet another material that has been investigated and reported, uses N,N,N′–tributylpentan–amidine embedded in polymer matrix of ethyl coating, working in conjunction with a SPR-based sensor [120]. CO_2_ ingresses into the polymer matrix and reacts with the network of N,N,N′ and polymer, and alters the electron density distribution and thus the permittivity leading to changes in the refractive index of the polymer network. Other coatings used are a polyallylamine–amino–carbon nanotubes, working with an exposed core of clad-etched FBG [121]; again, these are all redox reactions. Other researchers have created a nanoporous metal−organic framework fabricated by growing a cobalt zeolitic imidazolate framework (C-MOF) using zinc nitrate and cobaltous nitrate hexahydrate on repeated process cycles [122]. The chemical selectivity of the optical fibre CMOF originates from the specific adsorption capability of the coating itself to absorb significant amounts of CO_2_ to alter the refractive index of the C-MOF coating. The metal–organic framework (MOF) coated single-mode optical fibre sensor has been fabricated using different chemical component, such as nanoporous copper benzene 1,3,5-tricarboxylate, which showed encouraging results [123]. There are large number of strategies to create a CO_2_-selective gas optical fibre sensor; what we have done here is to show the diverse choices in detecting CO_2_.

### 3.3. Nitrous Oxides

The amount of nitrous oxides (NOs) in the environment along with other greenhouse gases has become an issue of global critical concern, with the publication of the Sixth Assessment Report by Intergovernmental Panel on Climate Change [124]. Therefore, the detection and accurate measurement of these gases has become a major concern. Whilst NOs are the third largest contributor to the greenhouse effect [125], it has a global warming potential 300 times that of CO_2_ over a 100-year time scale due to its longevity [126]. Additionally, NOs is the largest stratospheric ozone-depleting substance and is projected to remain so for the remainder of this century [127]. The detection of NOs can be a challenging task. Despite it being a thermodynamically potent oxidant, it is kinetically very stable and therefore behaves as an inert molecule.

Spectroscopic detection techniques have been applied using cavities formed in hollow core fibres, with absorption lines at wavelengths of 5.2 μm, which is anti-resonant [128], or using lines ~1522 nm [129] or 4.53 μm [130]. NOs detection suffers from the same issues as CO_2_ and CH_4_, as discussed previously. Some progress has been made by using materials to produce chemically selective sensors. For example, a thin polymer film of dimethylpolysiloxane/divinylbenzene, where the ingression of the nitrous oxide gas in the polymer reacts and creates a change in the refractive index, is then detected [131]. Another approach is using CNTs working in conjunction with polyethyleneimine polymer (PEI) (a nanomaterial) and surface plasmons. This optical fibre plasmonic sensor directly monitors the fluctuations in the refractive index resulting from the chemical reaction of the polymer with the gas and the CNTs. The addition of PEI creates an excess of electrons on the CNTs’ sidewalls. This creates efficient electron transport from CNT-PEI to N_2_O, thus changing the local permittivity (effective refractive index), amplifying the spectral response of the surface plasmon, which again is a redox reaction [132]. Other materials strategies are also used, such as the fluorescence of carbon quantum dots with surfaces functionalised with o-phenylenediamine. The o-phenylenediamine molecules that have adhered to the carbon quantum dot react with the NOs to form an electron-free triazole structure. This structure decreases the fluorescence of the CQDs. More NOs leads to a further decrease in the fluorescence of the CQDs [133]. In addition, another strategy is the use of calixarenes, which are building blocks in supramolecular receptors and can be functionalised by alkylation (substituting an alkyl group into the calixarene) [134]. In this case, alkylation was of tetrahydroxycalix [4] arene with sodium hydride and alkyl bromide for specific reaction with NOs. In the presence of NO_2_, alkylated calixarenes form stable calixarene-NO+ (nitrosonium) complexes that have a deep purple color, which is then used as the basis for fiber optic-based colorimetrics [134].

### 3.4. Water Vapour

Evidence shows that water vapour is a greenhouse gas and a major contributor to global warming. Water vapour absorbs energy in the far part of the infrared spectrum, indicating climate deterioration [135]. Data have been given that water vapour is responsible for a strong positive climate feedback, meaning that an increase in water vapour will produce an increase in the average global temperature; a higher water vapour concentration strengthens the greenhouse effect [136,137]. Furthermore, it has been shown that human activities can directly influence water vapour abundance in the atmosphere [138,139]. Generation of water vapour is very widespread, and occurs as a result of fossil fuel and biomass combustion, aviation, and evaporation due to irrigation for food production [140]. The development of water vapour sensors with remote sensing capabilities has become a major driving force for researchers and a myriad of articles have been published covering this need.

Various wavelength absorption lines/bands exists for water vapour in the infrared and near-infrared spectrum, at ~700 nm, ~935 nm, ~1392 nm, ~1800 nm, and ~2682 nm but there is some overlap with other greenhouse gases, such as CO_2_ and methane [141,142]. There are issues using longer wavelengths, such as the ~1.8 μm region and conventional fibre (made from SiO_2_) has higher attenuation coefficients. There are other alternatives, such as glassy materials, that can be used, and the chalcogenide fibres have been previously mentioned [114] though they have their own issues. Furthermore, light sources for the intracavity absorption spectra of fibre sensors are generally more expensive and less available, such as thulium-doped fibre lasers, and are tunable between 1.70 to 1.98 μm [143]. These issues have led researchers to devote their efforts to exploiting materials that react with water vapour. The authors of this review decided not to include coverage of this area, as there are excellent reviews and publications already available [144,145,146] but will briefly mention current research in this area. The focus has been on materials that react to the presence of water vapour. Polymers used in film coatings, such as polyallyamine hydrochloride (PAH), in conjunction with silica nanoparticles, which are hydrophilic, is an example. Absorption by the water molecules causes a refractive index change and hence a change in the reflectivity spectrum [147]. Other examples are hydrophilic gelatin [148], cobalt chloride hydrate [149], and graphene oxide [150]. Other strategies involve chitosan, a polysaccharide, which is a hydroscopic polymer, meaning that it causes swelling when in contact with water, and is measurable using a FBG [151]. Yet another technique is the use of a metal–organic framework [152]; a metal coordinated with organic ligands, which have mesoscale pores that absorb gas molecules. For example, copper-benzene 1,3,5-tricarboxylate (Cu-BTC). The colour of Cu-BTC changes in both depth and tone with water absorption when illuminated with a 468 nm light source, and allows the absorption spectra to be measured [152].

### 3.5. Other Trace Gases

There are myriad gas species/analytes that are of interest to researchers from different sectors of industry.

Hydrogen sulphide, which has importance in agricultural applications to slow down the ripening and deterioration of foodstuffs during storage, with typical concentrations of 1–80 ppm [153], along with obvious associated industries, oil and gas refining, plus mining, in which H_2_S is produced or used. Materials used for the detection of H_2_S vary from ZnO thin films with an outer layer of ZnO nanoparticle and sliver [154]. The H_2_S reacts with ZnO to form a zinc sulphide layer that covers the surface of the sensor, which effectively increases the refractive index of the medium on top of the ZnO coating. Another material used to detect H_2_S is a graphene oxide coating that produces a redox reaction [155]; other strategies can be found elsewhere [156].

Hydrogen gas sensing is becoming increasingly important as an alternative fuel [157,158], as gas needs to be stored securely and due to its explosive nature when mixed with air; leakage monitoring needs to be intrinsically safe, non-spark sensing schemes and that are portable and cheap. Optical fibre sensing of hydrogen has become an active area of photonic research. There is a major body of work on this particular subject with many very good reviews, the authors do not intend to repeat those works, but will point to them [159,160,161].

The favored detecting strategy is the use of palladium, which, on exposure to molecular H_2_, dissociates to be become 2H on the surface of Pd and diffuse into the Pd, creating palladium-hydride, and this leads to crystallographic and lattice changes in the Pd. These changes lead to the physical expansion of the Pd itself and thus the volume density of free electrons consequently decreases—which causes a reduction of both the real and imaginary parts of the Pd complex refractive index [162,163]. Therefore, FBG can be used to measure the strain and thus the H_2_ concentration or LPG to measure changes in the refractive index; there are issues with this approach, which can be found in the literature [159,160,161,162,163,164]. Other material transducers used to detect H_2_ are tungsten oxide (WO_3_) doped with a catalyst, which when utilizing a redox reaction changes the optical properties, such as reflectance, transmittance, absorption, and refractive index, of the WO_3_ [161].

Another significant gas is ammonia (NH_3_), and researchers have made considerable efforts to research and develop optical fibre sensors for the detection of ammonia. The reason for this substantial attention is that NH_3_ is amongst the most common chemicals manufactured and used across many different industries and are used for nitrogen-based fertilizer, pharmaceuticals, cleaning products, explosives, and in refrigeration [165,166,167]. Furthermore, ammonia is a cause of great environmental pollution [168]; atmospheric ammonia makes significant contributions to large-scale nitrogen eutrophication and acidification of ecosystems. Recently, there are been interest in using ammonia-sensitive dyes, such as tetraphenylporphyrin tetrasulfonic acid hydrate (TPPS), which exhibit absorption spectra of ~480 nm and ~700 nm in the presence of NH_3_ and are used with microstructured optical fibers [169]. Another approach of interest is the use of nanostructured materials like Ag/ZnO composite nanostructures that have redox reactions and is interrogated using SERS [170], or nanocrystalline samarium oxide (Sm_2_O_3_) with an absorption spectra of ~610 nm [171]. Another material to find use in the detection of NH_3_ is graphene oxides; using graphene oxide and cellulose acetate [172], adhesion of HN_3_ to the surface changes the refractive index of LPGs GO-coated tapered optical microfiber. Alternatively, the absorbance characteristics of GO at ~550 nm and ~750 nm can be used to detect the presence of HN_3_ [173].

In this section, we deal with volatile organic compounds (VOCs). MOS sensors have been used to detect hydrocarbon alcohol vapours, such as methanol, ethanol, and propanone [174], similar to the previously mentioned gases. There are, however, issues in this application which caused researchers to resort to employing statistical techniques, such as principle component analysis [174,175], in order to identify target gases [174,175]. Various reactive materials have been used to work in conjunction with an optical fibre sensing mechanism, such as Ag/graphene/Ti_3_C_2_T_x_, where T represents a surface functional group working as a potential SPR sensor for detecting propanone [176], yielding sensitivities of ~5000 nm/RIU. Researchers have investigated many materials and different sensing strategies in this area; for example, polydimethylsiloxane working with non-adiabatic micro-nano fiber formed in the mid-section of an optical fibre biconical taper, based on an evanescent field [177] has shown promising results in the case of ethyl alcohol (ethanol) for a concentration range of 0–140 ppm. Researchers have also shown interest in other VOCs, such as composite materials using absorption spectra, such as the aromatic organic compounds toluene and xylene, as well as cadmium arachidate (CdA) with single-walled carbon nanotube (SWCNT) composites materials and the absorption spectra to detect these aromatic organic compounds [178]. The authors would like to inform readers that this area of research merits its own review. We direct interested readers to References [179,180].

## 4. Comparison of Performances

There is a broad range of characteristics available in the numerous publications quoted in this review, such as interrogation schemes, sensitivity, limit of detection (LOD), use, and response times. All but the interrogation schemes are self-evident. We have included interrogation schemes because they give the reader some indication of costs, complexity, and equipment needed. This information plays an important role in deciding a specific strategy. Furthermore, a comparison needs to be made amongst the optical fibre sensors and an accepted conventional sensing scheme, such as spectroscopic techniques. Due to shortcomings in conventional gas sensing [181], there has been growing interest in metal–oxide semiconductors (MOS) [182,183,184,185]. MOS detection schemes show distinct advantages over more conventional spectroscopic techniques, such as lower fabrication costs, miniaturisation, integration and multiplexing capabilities. On the other hand, the weakness of MOS sensing schemes is an issue related to stability and chemical selectivity, along with the need to operate above ambient temperature. In addition, there is the problem of spark hazards, because their electrical operation poses serious issues in some specific environments and for a remote sensing due to electrical power requirements. Nevertheless, their use is widely accepted as a conventional sensing approach.

Table 1 lists the performance details of optical fibre devices used in the detection of methane. It can be noted that these devices operate in the concentration regime, from zero to a few percent, except for [99,100]. The LOD of these sensors approaches the concentration levels in the atmosphere, which is 0.00017%. Remote sensing capability has yet to be achieved, but is being improved for use as a remote sensor in a natural environment. Whilst the concentration detection levels of MOS sensors are typically ~0.2% operating at ambient temperatures, their LOD needs to be improved to 0.005%, but requires an elevated temperature operation and, therefore, increased electrical power is needed for remote operating systems [184]. MOS sensors are used for methane leak detection and monitoring in chemical processing. Concentration levels of 5 to 15% in air are potentially explosive [185]. MOS sensors are electrically based and thus have the potential risk to ignite methane causing optical sensors [99,100,102,104] to have a distinct advantage. Spectroscopic sensors [93] are more lab-based sensors. The use graphene coated based sensors are showing promise with regard to remote sensing capabilities.

In further consideration of methane sensors, SPR sensors using graphene and PMMA indicate a spectral sensitivity of 10^3^ nm/% and a LOD of 7 × 10^−4^% [106], although over a limited range, and they have applicability in atmospheric methane measurement. In the case of general industrial applications, a more extensive range of operation is necessary, with faster response times, for controlling processes. Pt/ZnO SPR devices appear to yield the greatest range of use with the fastest response and an ability to regenerate, whilst cavities and crytophane [102,103,105] yield better LODs, thought the response times are too slow for process control. The spectroscopic technique has a reasonable response time but implementation away from a laboratory environment becomes more challenging with the use of non-conventional optical fibres, such as chalcogenide fibres.

Table 2 provides a summary of optical fibre sensors for carbon dioxide detection, a gas that is a leading member of the greenhouse gas family and is therefore of global concern because of its impact on the Earth’s eco-system [124]. Carbon dioxide’s atmospheric concentration is ~0.04%, and has been increasing over the last 50 years from ~0.032% [186]. Thus, environmental sensors need to detect changes in the atmosphere of ~0.0001%. A number of MOS devices have LODs from 0.02% to 0.1%, but require 100 s of degrees centigrade to operate [183], which is an issue for sustained periods of standalone remote sensing. Recent published data on MOS sensors suggest ~0.2% to 1%, though still requiring elevated temperatures [112,113,116,123,181]. All optical fibre devices operate at room temperature. Spectroscopic sensors offer good performance, but are laboratory based. Spectroscopic sensors have good performances, but are very much lab-based schemes with the potential to be used in-situ [116,181] as well as for environmental applications and [115,117,123] with largest range of operation, but they still have LOD and resolution problems. 

Scrutinising the performances and comparing them to each other, spectroscopic sensors using cavities, either PCF or intra-cavity, yield the best LODs [112,113] but have slow response times, although this is not critical for atmospheric measurement of CO_2_. The reaction and absorption spectra given by NiO/rGO coatings [116] yield similar results to spectroscopic sensors but with a faster response over the same concentration ranges. Another point of concern with cavity sensors [112,113,187,188] is that they need additional apparatus to deliver the gas sample into the cavities, either via vacuum or high pressure, which is not needed for reactive coatings, such as NiO/rGO [116] or FOM [123,189]. Furthermore, for industrial monitoring applications of control processes, all sensors operate over a wider range, and have relatively slow response times [114,115,117,122,123]. LSPs working in conjunction with carbon-nanotubes yields the fastest response of 12 s.

Table 3 gives a summary of nitrous oxide optical fibre sensors, which as previously mentioned, is the third largest contributor to the greenhouse effect [125] due to it having a global warming potential 300 times that of CO_2._

Nitrous oxides are also a constituent of “smog”, made famous in the 1950s in London, but which is still seen in some major cities today. Photochemical smog, apart from the haze it produces, is irritating to the eyes and damages plant life. Nitrous oxides are quite detectable by smell but, at concentrations of 4 ppm (~4 × 10^−4^%), anaesthetises the nose creating the possibility that increased concentrations in an environment may go unnoticed, causing potential health risks. The nitrous oxide concentration level in the atmosphere is approximately ~0.00003% and has been steadily increasing over the last three decades at a rate of ~10^−5^%/year. MOS sensors based on SnO, ZnO, WO_3_, In_2_O_3_, and CuO yield LOD of ~5 × 10^−4^% but require elevated temperatures for operation and suffer from cross-sensitivity with other compounds [183]. In general, optical fibre devices outperform MOS devices in this application, particularly concerning chemical selectivity and ignoring spectroscopic sensors (which are predominantly laboratory-based in nature).

The sensors showing potential [131,133,134] are able to achieve the desired performance for environmental measurements without elevated temperatures and provide greater flexibility for other industrial applications with larger ranges of operation [132]. Spectroscopic sensors [128,130,133,190] all yield LOD and concentration ranges suitable for the detection of atmospheric nitrous oxides and while response times are not critical for environmental sensing, the measuring apparatus is laboratory-based. The only large concentration range of operation for possible industrial applications is given by LSP and carbon-nanotubes [132], but the response times are still slow at 19 s. It appears that the detection of nitrous oxides with optical fibre sensors has still some way to go.

Table 4 gives a summary of water vapour optical fibre sensors. The amount of water vapour in the atmosphere can vary from trace to ~4%, and is increasing due to human industrial and agricultural activities [138,139], by about 1–2% per decade [124]; thus ~0.1% per year. Therefore, for standalone and remote environmental monitoring, LOD and resolution needs to be on the order ~0.01% or better. MOS sensors do react with water vapour, but suffer cross-sensitivity with other compounds [145]. Inspecting Table 4, there are potential candidates for atmospheric detection of vapour applications [152,191,192,193] with approximately the correct LOD and concentration ranges, with the additional advantage of low cross-sensitivity with other substances. 

This group uses reactive coatings, such as FOM [152] (~0.0025%), graphene [195] (~0.05%), and silica xerogel/gelatin [192] (~0.3%) along with a spectroscopic sensor working with methylene blue [191] (~0.062%). Considering other applications, the response potential is important and the best performing option is the taper [93] and LPG [194], although there is concern regarding selective water vapour spectral response with the taper [195] in the case of measuring the refractive index.

A more conventional approach is to exploit chemiresistive devices, for example, those that use multi-walled carbon nanotubes and functional polymers. These devices have produced good results, reasonable resolutions and LOD [196,197] and work at ambient temperature. Chemiresistive sensors have a major drawback which is the LOD, which is not sufficient for the low percentage range of concentrations in the atmosphere and neither is the resolution [145,196,197].

Here we have made comparisons for a given gas species and compared sensor performances with each other and their potential applications. Trying to perform a comparison for different gases and different classes of sensors is problematic with regards sensitivity, selective spectral response, etc. What the authors aim to give is an overall impression of the use for each class of sensor. Inspecting the performances of the all the sensors, the reactive coatings working in conjunction with the evanescent fields or the plasmonic mechanism appear to have far greater potential to achieve the desired performance specifications, as well as to be portable and to be used as standalone remote sensing schemes. The reason for this is the minimal amount of peripheral equipment needed to make the measurements, and thus the electrical power budget is less demanding. Material scientists and chemists are addressing the issue of chemically selective spectral response, with a myriad of materials being investigated to address the problem.

Furthermore, the authors would like to raise the issue of chemical selectivity. This particular property of optical fibre gas sensors arises from the physical response of a reactive coating with the target gas analyte. Although this is difficult to demonstrate, it usually happens when there is another gas, chemically similar to the target gas, present. To quantify the chemical selectivity of the sensor, the maximum response to the target gas analyte is taken as a ratio of the maximum response for a non-target gas, [***R_n_***]. The authors accept that [***R_T_***] is a crude measure, but this can be thought akin to probability; therefore, with a [***R_n_***] value very small compared to [***R_T_***] $\mu_{p}=\frac{\left[R_{n}\right]}{\left[R_{T}\right]+\left[R_{n}\right]}\to 0$ and so there is a low possibility of misreading. If [***R_n_***] = [***R_T_***], then ***μ_p_*** = 0.5, thus yielding an equal probability of reading the sensor correctly. Spectroscopic sensors are more straightforward; an absorption line is excited at a specific wavelength and can be attributed to a specific gas. They do, however, have some practical problems to overcame, for example, remote standalone sensing capabilities, as already highlighted.

The chemical selectivity of optical fibre gas sensors, already discussed in this review, is shown in Table 5. Inspection of Table 5 indicates that many of the researchers did not include this in their published work. There could be several reasons for this omission, the assumption of common knowledge, oversight, or assumptions on the specific environment the sensors were being used in (therefore no experiments on selectivity needed). In addition, water vapour chemical selectivity was not demonstrated, this is probably due the significant index changes for water vapour (changes of approximately ~|1.5 × 10^−6^| at room temperature [198]) compared to realistic refractive index changes due to a single atmospheric gas species. The possibility of misinterpretation, especially for nitrous oxide is also possible. Improvement occurs for carbon dioxide and more so for methane; the two results that gives high confidence that they are true measurements is [99], which is a plasmonic based sensor for methane and MOF (cobalt zeolitic imidazolate), based on absorption spectroscopy [122], at ~5% and ~7% respectively probability of miss-interpretation. Finally, Table 5 shows that the spectral response with chemical selectivity using reactive coatings sensor still needs further research.

## 5. Concluding Remarks

The authors of this review hope that they have provided a comprehensive appraisal of the “state-of-the-art” of optical fibre sensing devices, and that they also provided insight into the various measuring techniques available, along with their advantages, disadvantages, and the pitfalls of other competing devices. The authors accept that there are many other gases that researchers will want to detect and quantify that are not been dealt with in this review. We sincerely hope that this review serves as a springboard for new researchers contemplating a career in optical fibre sensing devices. During the writing of this review, The Intergovernmental Panel on Climate Change published their 2021 report on global warming with the warning that we need to reduce greenhouse emissions. With this in mind, we have concentrated on the detection and measurement of major greenhouse gases and the challenges that still exist in their detection. We hope this helps in some small way in the fight that is coming to us all in the near future.

## Figures and Tables

**Figure 1 sensors-21-06755-f001:**
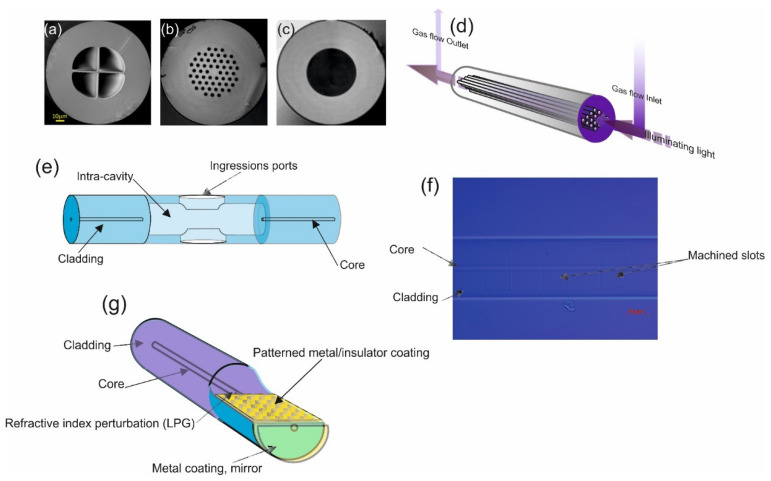
Typical schematics of various spectroscopic optical fibre sensors. Various images of differing types of photonic crystal optical fibres: (**a**) grapefruit, (**b**) large mode area photonic crystal fiber, (**c**) hollow optical fibre; capillary fibre. (**d**) Schematic of the operational use of photonic crystal optical fibre as a gas sensor. (**e**) Schematic of intra-cavity optical fibre for gas sensing, which would ulitise the capillary fibre. (**f**) A microscope image of femto-second laser micro-machined “slot” cavities close to the core of an optical fibre. (**g**) Schematic of optical fibre surface-enhanced Raman spectroscopy (SERS) sensor. SERS uses a nano-patterned material like Au or Ag nano-particles/colloid. Figure 1a is reproduced from D. Lopez-Torres et al., “Optical fiber sensors based on microstructured optical fibers to detect gases and volatile organic compounds—a review”. Sensors 20, no. 9 (2020): 2555.

**Figure 2 sensors-21-06755-f002:**
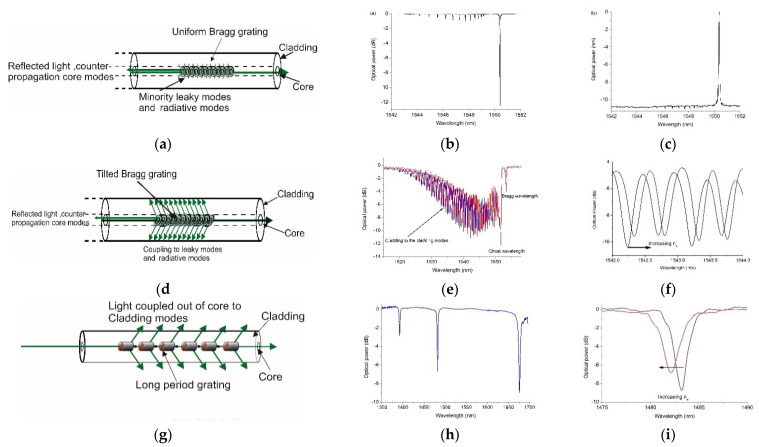
The three types of fibre grating sensors and their spectral response. (**a**) Schematic of fibre Bragg grating. (**b**,**c**) A typical fibre Bragg grating spectral response in reflection and transmission. (**d**) Schematic of tilted fibre Bragg grating. (**e**,**f**) A typical tilted fibre Bragg grating spectral response in transmission and the response to change in the surrounding medium index. (**g**) Schematic of long period grating. (**h**,**i**) Spectral responses in transmission and the response to change in the surrounding mediums index.

**Figure 3 sensors-21-06755-f003:**
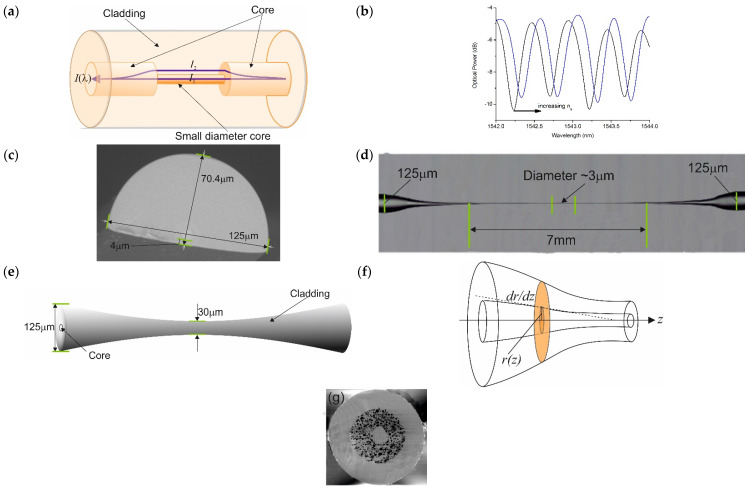
Different types of evanescent field optical fibre sensors. (**a**) Schematic of an in-fiber Mach-Zehnders using a section of smaller-diameter core fusion spliced into an optical fibre. (**b**) A typical transmission spectrum for an in-fiber Mach-Zehnders and its spectral response to a change in the surrounding environments refractive index. (**c**) An image of a section of D-shaped optical fibre with the core/cladding interface to the flat part of the D is 4 μm. (**d**) A side view image of biconical tapered optical fibre with a central section of nano/micro fibre, with a diameter of 3 μm. (**e**) Schematic of typical biconial taper with a central waist of 30 μm. (**f**) A taper showing the parameters of interest to determine spectral behaviour. (**g**) An image of a cross-section of a random-hole optical fibre.

**Figure 4 sensors-21-06755-f004:**
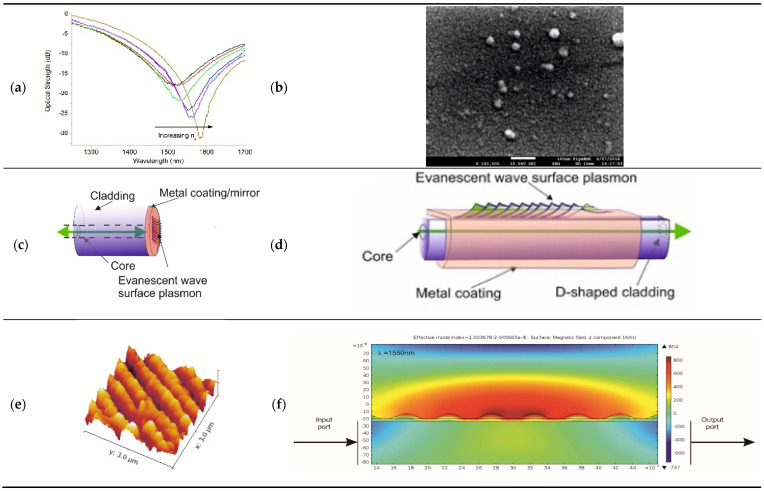
(**a**) A typical spectral response optical fibre SPR sensor with increasing index of surrounding medium, in this case the metal coating is Ag with a thickness of 28 nm; a D-shaped SMF208 fibre. (**b**) An example of a coating with metallic surface artefacts, a high-resolution image of the ZnO and Pt coating surface showing distribution of the nucleation surface features. (**c**,**d**) Two examples of different optical fibre plasmonic configurations for conventional SPs and LSPs. (**e**) An atomic force microscopic image of a nano-antennae array with Au on the apexes of each nano-antenna. (**f**) FEM numerical modelling of a section of the nano-antennae array generated by conjoined LSPs. The z-component of the magnetic field at a wavelength of 1550 nm with a surrounding RI of 1.36, the physical dimensions in the model come from atomic force microscopy measurements.

**Table 1 sensors-21-06755-t001:** Summary of various optical fibre sensors for the detection of methane.

Material for Selective Sensing	Interrogation and Mechanism	Sensitivity	L. O. D.	Range	Comments
		nm/%	%	%	
─	Spectroscopy, ~1.6 and ~3.2 μm, cavity and PCF	─	~0.007	0–0.1	[93] regeneration, response time 10 s
Pt/ZnO	OSA, SPR, ~1420 nm	0.01	~0.15	0–100	[99] regeneration, response time 1 s
SnO_2_/Graphene	OSA, absorption, ~1570 nm	─	~1.0	0–55	[100], no more Info
Crytophane E	OSA, absorption, cavity ~1550 nm	−1.6	~0.06	0–5	[102], response time 5 min
Crytophane E	OSA, absorption, PCF cavity ~1350 nm	4.6	~0.04	0–3	[103] regeneration, response time 60 s and 180 s
Crytophane E	OSA, absorption, PCF ZM ~1550 nm	1.272	~0.02	0–5	[104], no more Info
Crytophane A	OSA, absorption, PCF cavity ~1350 nm	−1.99	~0.2	0–3.5	[105], no more Info
Graphene+CNTs and PMMA	OSA, absorption, SPR ~450 nm and ~720 nm	1000	~0.0007	0–0.01	[106], no more Info
Graphene+Ag	OSA, index, SPR and LPG ~1538 nm	0.34	~0.1	0–3.5	[107] regeneration, response time 50 s and 160 s
Crytophane A	OSA, index, LPG ~1545 nm	2.5	~0.2	0–3.5	[109] regeneration, response time 60 s and 180 s
Crytophane A	OSA, index, LPG ~1550 nm	6.39	~0.015	0–3.5	[110], no more Info

**Table 2 sensors-21-06755-t002:** Summary of various optical fibre sensors for the detection of carbon dioxide.

Material for Selective Sensing	Interrogation and Mechanism	Sensitivity	L. O. D.	Range	Comments
		/%	%	%	
─	Spectroscopy, ~3.95 and ~4.23 μm, cavity, chalcogenide fibre	─	~0.02	0–1.9	[111] regeneration, response time 5 s
─	spectrometer, absorption, PCF cavity ~1530 nm	─	~0.0005	0–0.05	[112] regeneration, response time 50 s
─	spectrometer, absorption, cavity ~2 μm	─	~0.0006	0–0.1	[113], no more Info
─	FTIR, absorption, chalcogenide fibre cavity ~4.2 μm	1 au	~0.5	0–100	[114] regeneration, response time 1 min
HPTS (8-Hydroxypyrene-1,3,6-trisulfonic acid trisodium salt hydrate) and Xerogel	absorption, ~396 nm and ~460 nm	0.3 dB	~0.03	0–30	[115] regeneration, response time 50 s and Regen 100 s
Nickel oxide and reduced graphene oxide	spectrometer, absorption, ~650 nm and 771 nm	1400 au	~0.0005	0–0.05	[116] regeneration, response time 16 s and Regen 22 s
Carbon nanotubes	OSA, absorption, LSP ~1432 nm and ~1544 nm	0.04 nm	~0.05	0–100	[117] regeneration, response time 10 s and Regen 12 s
TB-PAM(tributylpentan-amidine) and ethyl cellulose	OSA, absorption, Au and polymer matrix ~650 nm	400 au	~0.01	0–0.03	[120] regeneration, response time 100 s
Carbon nanotubes and Polyallylamine	OSA, FBG ~1550 nm	0.1 nm	~0.01	0.1–0.4	[121], no more Info
Metal-Organic Framework Cobalt Zeolitic imidazolate	Spectrometer ~200 nm to ~650 nm absorption	1.1 au	~0.5	0–100	[122] regeneration, response time 20 s and Regen 80 s
Metal-Organic Framework Copper benzne 1,3,5-tricarboxylate	OSA ~1570 nm absorption	0.03 au	~0.002	0–100	[123] regeneration, response time 10 s and Regen 50 s
SnS2 (Tin disulphide)	OSA ~445 nm emission	8 nm	~0.008	0–0.06	[187], no more Info
─	Spectroscopic hollow optical fibre, cavity ~4.28 μm and 4.25 μm	─	~0.02	0–4	[188], no more Info
Metal-Organic Framework HKUST-1(benzene-1,3,5-tricarboxylate)	OSA ~700 nm LPG	25 nm	~0.04	0.05–4	[189] regeneration, response time 60 s and Regen 5 min

**Table 3 sensors-21-06755-t003:** Summary of various optical fibre sensors for the detection of nitrous oxides.

Material for Selective Sensing	Interrogation and Mechanism	Sensitivity	L. O. D.	Range	Comments
		/%	%	%	
─	Spectroscopy, ~5.26 μm, cavity, hallow optical fibre	−0.634 v	~0.000014	0–0.0001	[128] regeneration, response time 70 s and Regen ~100 s
─	Spectroscopy absorption, ~1552 nm, cavity,	3000 dB	~0.00005	0–0.001	[129] regeneration, response time 36 s
─	Spectroscopy, ~4.53 μm, cavity, hallow optical fibre	5 × 10^5^ au	~0.00001	0–0.0003	[130] regeneration, response time 23 s and Regen 30 s
Divinylbenzne and siloxone ploymer	spectrometer, SPR ~1550 nm	5 × 10^7^ dB	~10^−7^	0–1.8 × 10^−6^	[131] regeneration, response time 5 s and Regen 360 s
Carbon nanotubes, Poly ethyleneimine, Au	OSA, absorption, LSP ~1450 nm	0.05 nm	~0.0109	0–100	[132] regeneration, response time 19 s and Regen 400 s
Carbon qunatum dots and o-phenylenediamine	spectroscopic, flourescence ~400 nm	10^5^ au	~0.00003	0–0.01	[133], no more Info
calix [4] arenes and O-hexyl	spectrophotometer, absorption ~582 nm	185 au	~0.0005	0.0007–0.0015	[134] regeneration, response time 2 min and Regen 20 min
─	Spectroscopy, ~1500 nm, cavity, hallow optical fibre	82 μV	~0.001	0–2.5	[190] regeneration, response time 60 s and Regen 6 min

**Table 4 sensors-21-06755-t004:** Summary of various optical fibre sensors for the detection of water vapour.

Material for Selective Sensing	Interrogation and Mechanism	Sensitivity	L. O. D.	Range	Comments
		/%	%	%	
─	Spectroscopic, ~1.9 μm, cavity	─	~10^−5^	0–0.1	[143] response time 0.5 ms
Poly allylamine hydrochloride and silica nano particles	Spectrometer reflective spectrum, ~500 nm and ~650 nm	0.07 nm	0.43	~50–95	[147] regeneration, response time 3.1 s and Regen 57.3 s
Gelatin	OSA, tranmission spectra, Taper ~1550 nm	−0.14 dB	~0.2	~9–94	[148] regeneration, response time 70 ms
Cabolt chloride ethylene oxide	OSA, LPG ~1550 nm	~−0.23 nm and ~0.33 nm	~0.1	~50–77 and 77–95	[149] regeneration, response time 20 s
Graphene oxide	OSA, Transmission spectra, Mach-Zhender; core-offset ~1550 nm	0.349 dB	~0.2	30–77	[150], no more Info
Chitosan (polysaccharide)	spectrometer reflective spectrum, FBG, 1541 nm	0.107 nm	~0.1	30–77	[151] regeneration, response time ~2 s and Regen 60 s
Metal Organic Framework; Copper benzene 1,3,5 tricarboxylate	spectrometer, absorption ~490 nm	0.15 au	~0.0025	0–1	[152] regeneration, response time 23 s and Regen 5 min
Sol-Gel doped with Methylene Blue	Spectroscopy, absorption ~540 nm	0.087 dB	~0.062	0–70	[191] regeneration, response time 20 s and Regen 3 min
Silica Xerogel, gelatin and tetraethylortho silicate	spectrometer reflective spectrum, 630–670 nm	~0.08	~0.3	4–100	[192] regeneration, response time 10 s and Regen 2 min
Graphene quantum dots	Scanning laser, transmission, cavity, 1550 nm	0.567 nm	~0.05	~11–85	[193] regeneration, response time 5.5 s and Regen 8.5 s
Tungsten dissulphide and Au	OSA transmission spectra, LPG and SPR, ~1535 nm	0.037 nm	~0.2	30–80	[194] regeneration, response time 0.3 s and Regen 1.5 s
─	OSA Reflective spectra, PCF/Taper Cavity ~1550 nm	−0.166 dB	~0.1	30–90	[195] regeneration, response time 0.4 s and Regen 0.2 s

**Table 5 sensors-21-06755-t005:** Summary of chemical selectivity of various optical fibre sensors.

Target Gas and Material for Selective Sensing	Gas Target Response	Non-Target Response	*μ_p_*	Comments
Methane Pd-Pt/ZnO (Spectroscopic)	~+80%	~+15%	0.158	REF [98]
Methane Pt/ZnO (SPR)	~+0.4 nm	~−0.02 nm	0.048	REF [99]
Methane graphene-carbon nanotubes-poly (methyl methacrylate (SPR)	~+30 nm	~+9.0 nm	0.231	REF [106]
Nitrous Oxide Pt/ZnO (SPR)	~+5.7 nm	~−4.0 nm	0.412	REF [132]
Nitrous Oxide carbon quantum dots (Spectroscopic)	~+0.34%	~+0.2%	0.370	REF [133]
Carbon Dioxide NiO/Reduced Graphene Oxide (Spectroscopic)	~+82%	~+41%	0.333	REF [116]
Carbon Dioxide carbon nanotubes (LSP)	~+4 nm	~+0.7 nm	0.149	REF [117]
Carbon Dioxide MOF (Absorption Spectroscopy)	~+41%	~+3%	0.068	REF [122]

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
