# Peer review of "A Review: Application and Implementation of Optic Fibre Sensors for Gas Detection"

_sensors, 2021, doi:10.3390/s21206755_

Round 1
Reviewer 1 Report
The authors reviewed and discussed the optical fiber gas sensing technologies in recent years. They started from the four different gas sensing mechanics and the corresponding fiber structures; the sensing and performance comparison for four typical gas angles. The optical fiber structures of spectral absorption, grating, evanescent field and plasmonics types should be compared and analyzed in a table, especially the related sensing performance (such as sensitivity and selectivity) should also be compared. In addition, please pay attention to the following problems.
PCF is well-known as Photonics Crystal Fiber instead of “poly-crystalline fibre” in this manuscript. Please check and fix it.
“The issue of using FBGs is there intrinsic insensitivity…” should be “their”
Author Response
Firstly, the authors would like to thank the reviewer for their time and effort for their careful review of the manuscript we are very much grateful to both reviewers for their valuable comments, which helped us to improve our manuscript. The answers to the reviewers' comments are given in the text below and the revised sections are highlighted correspondingly in the manuscript.
General comments:
The comments made by the Reviewer 1 with regards making an overall comparison, we originally was going to do this but after careful consideration, it was not the correct choice of presentation. Please let us explain, in the paper we are looking gas sensors for various target gas analytes with a diverse range of sensing strategies were the researchers in their own projects are only concerned with the target gas/gases. Thus comparing the performance of the water vapour sensors to the methane sensor is doesn’t make sense because the target analytes are chemically and optically different. Therefore, comparisons within each group makes for a more meaningful and logically approach.
Saying this the Reviewer made very valid point and we have expanded the comparison between the optical fibre sensors within each gas species group. Furthermore, towards the end of the review paper we have given over a section of the paper to a general overview comparison and observation of all the sensors and added chemical spectral selectivity for the various sensors that demonstrated this property. Along with a simplistic way to evaluate this property for all the sensors.
The comments made by the Reviewer 2 on the inclusion of the VOCs sensors is very good point and an oversight by the authors. We have now refereed to the class of gas sensors and added a section and seven new references.
Specific Responses
Academic Editor
The Authors must include a brief description of the prior art on the Fiber-optic gas-phase biosensors for VOCs
Yes, also commented by Reviewer, this was an oversight by both authors and has been corrected, please see the responses to Reviewer 2 and general comments.
Reviewer One
Comment 1
The authors reviewed and discussed the optical fiber gas sensing technologies in recent years. They started from the four different gas sensing mechanics and the corresponding fiber structures; the sensing and performance comparison for four typical gas angles. The optical fiber structures of spectral absorption, grating, evanescent field and plasmonics types should be compared and analysed in a table, especially the related sensing performance (such as sensitivity and selectivity) should also be compared.
Ans 1(a)
New passages of text and discussion has been added to each gas in the section 4 of the paper “Comparison of performances” relating to the comparison of the performance of each gas sensors. Also please see general comments
For Methane page 16, paragraph 3 and 4, lines 652 to 668
For Carbon Dioxide page 17/18, paragraph 2 and 1, lines 682 to 693
For Nitrous Oxide page 20, paragraph 2, lines 710 to 717
For Water Vapour page 21, paragraph 2, lines 733 to 740
Ans 1(b)
New passages of text on an overview of the sensors performances is given across all sensors, along with discussion and a new table 5 on the property of chemical selectivity.
Page 21, paragraphs 2, 3, 4 5 and 6. Lines 733 to 774
Page 22, paragraphs 1. Lines 775 to 778
Page 23 Table 5 “Summary of chemical selectivity of various optical fibre sensors”
Added new reference
[198] Mathar, Richard J. "Refractive index of humid air in the infrared: model fits." Journal of Optics A: Pure and Applied Optics 9, no. 5 (2007): 470
Comments 2
PCF is well-known as Photonics Crystal Fiber instead of “poly-crystalline fibre” in this manuscript. Please check and fix it.
Ans 1(c)
Done, page2, line 75
page 3, line 100
page 4, line 129 and 130/131
Comments 3
“The issue of using FBGs is there intrinsic insensitivity…” should be “their”
Done page 4, changed to “The issue of using FBGs is they are intrinsically”, line 166
Reviewer Two
Thank you for sharing author’s knowledges to the global society. I believe that the manuscript of this review article is very well-written. I wanted to comment only one thing. Fiber-optic gas-phase biosensors for VOCs were studied in terms of selective gas sensing. Authors should mention those sensors in the manuscript. Totally, quite positive to accept for publication.
Ans 2(a)
A new passage of text and reference have been to paper, a brief discussion on the performance of VOCs sensor and the materials that can be used and added 7 new references. Also please see general comments.
Page 15, in the section entitled “Other Trace Gases” paragraph 6, lines 613 to 617
Page 16, in the section entitled “Other Trace Gases” paragraph 1, lines 618 to 628
Added new references
174 Yan, Meng, Yi Wu, Zhongqiu Hua, Ning Lu, Wentao Sun, Jinbao Zhang, and Shurui Fan. "Humidity compensation based on power-law response for MOS sensors to VOCs." Sensors and Actuators B: Chemical 334 (2021): 129601.
175 Tharwat, Alaa. "Independent component analysis: An introduction." Applied Computing and Informatics (2020).
176 Sudheer, V. R., SR Sarath Kumar, and S. Sankararaman. "Ultrahigh Sensitivity Surface Plasmon Resonance–Based Fiber-Optic Sensors Using Metal-Graphene Layers with Ti 3 C 2 T x MXene Overlayers." Plasmonics 15, no. 2 (2020): 457-466.
177 Shi, Bufan, Yang Sun, Wanlu Zheng, Naisi Zhu, and Ya-nan Zhang. "Highly-sensitive ethanol gas sensor based on poly dimethylsiloxane coated micro-nano fiber." In 2020 Chinese Control And Decision Conference (CCDC), pp. 4959-4962. IEEE, 2020.
178 Consales, M., A. Crescitelli, M. Penza, P. Aversa, P. Delli Veneri, M. Giordano, and A. Cusano. "SWCNT nano-composite optical sensors for VOC and gas trace detection." Sensors and Actuators B: Chemical 138, no. 1 (2009): 351-361.
179 Elosua, Cesar, Ignacio R. Matias, Candido Bariain, and Francisco J. Arregui. "Volatile organic compound optical fiber sensors: A review." Sensors 6, no. 11 (2006): 1440-1465.
180 Pawar, Dnyandeo, and Sangeeta N. Kale. "A review on nanomaterial-modified optical fiber sensors for gases, vapors and ions." Microchimica Acta 186, no. 4 (2019): 1-34.
Yours Sincerely
Thomas Allsop and Ron Neal
Reviewer 2 Report
Thank you for sharing author’s knowledges to the global society. I believe that the manuscript of this review article is very well-written. I wanted to comment only one thing. Fiber-optic gas-phase biosensors for VOCs were studied in terms of selective gas sensing. Authors should mention those sensors in the manuscript. Totally, quite positive to accept for publication.
Author Response

(The authors gave the same response as above.)

Round 2
Reviewer 2 Report
I thank to authors who answered my comments.